# Size Effects in Internal Friction of Nanocrystalline Aluminum Films

**DOI:** 10.3390/ma14123401

**Published:** 2021-06-19

**Authors:** Nhat Minh Dang, Zhao-Ying Wang, Yun-Chia Chou, Tra Anh Khoa Nguyen, Thien Ngon Dang, Ming-Tzer Lin

**Affiliations:** 1Graduate Institute of Precision Engineering, National Chung Hsing University, Taichung 40749, Taiwan; D108067005@mail.nchu.edu.tw (N.M.D.); D108067002@mail.nchu.edu.tw (Z.-Y.W.); khoa.eur@gmail.com (T.A.K.N.); 2Siliconware Precision Industries Co., Ltd., Taichung 427, Taiwan; bj611221@gmail.com; 3Faculty of Mechanical Engineering, Ho Chi Minh City University of Technology and Education, Ho Chi Minh 700000, Vietnam; ngondt@hcmute.edu.vn; 4I-Center for Advanced Science Technology, National Chung Hsing University, Taichung 40749, Taiwan

**Keywords:** ultra-thin al films, energy loss, internal friction, thickness dependence, quasi-static properties of thin film

## Abstract

Al thin film is extensively used in micro-electromechanical systems (MEMS) and electronic interconnections; however, most previous research has concentrated on their quasi-static properties and applied their designs on larger scales. The present study designed a paddle-like cantilever specimen with metal films deposited on the upper surface to investigate the quasi-static properties of Al thin film at room temperature under high vacuum conditions at microscopic scales. Energy loss was determined using a decay technique in the oscillation amplitude of a vibrating structure following resonant excitation. Grain size and film thickness size were strictly controlled considering the quasi-static properties of the films. This study found that the internal friction of ultra-thin and thin Al films was more dependent on the grain boundaries than film thickness.

## 1. Introduction

Al thin films are popularly utilized for microelectronics manufacturing and packaging. Al thin films used in integrated circuits (ICs) and micro-electromechanical system (MEMS) structures are superimposed on top of layer upon layer and are often directly connected to each other. During fabrication, different temperatures correspond to the process; this means that the whole structure is affected by fluctuations in temperature during each process. The temperature can be increased or decreased during treatment because the coefficients of thermal expansion between the contact layers become uneven and they create mechanical stress fluctuations. When these stresses are greater than the yield stress of the film or become too large [1,2,3,4], mechanical damage can occur, which could lead to failure.

In MEMS, there are moving components that form the system, and these moving components are subjected to dynamic loads by the metal films deposited on the components. Operating frequencies in the GHz or MHz range are often used in miniature structures and bulk material properties, but they are not suitable for designs operating at excessively high frequencies. As a result, the dynamic properties of thin metal films, particularly energy loss, are critically important in their application.

MEMS devices are electrically controlled, but they are susceptible to mechanical failure and damping due to internal friction that is created by the impact between thin metal films on dynamic loads. At a micrometer scale or nanometer scale, measuring the internal friction of a thin film could help to determine the relaxation processes involved in boundary diffusion and sliding [5,6,7]. Prieler et al. [8] and Illés et al. [9] performed systematic investigations of the relaxation of thin Al films on a Si substrate, in which the grain size was completely controlled and varied solely with film thickness [8,9]. The results showed that the grain boundary sliding and diffusion were related to internal friction. In a different study, the internal friction between free-standing membranes and films attached to a substrate was compared [10,11]. The authors concluded that, because the substrate restricted the grain boundary sliding near the interface, the internal friction in free-standing films was greater than that in the film attached to a substrate.

In a recent study, Fujiwara et al. [12] found that internal friction in thin films was due to anelastic grain boundary processes. Nishino [13,14,15] illustrated that the dependence of amplitude on the internal friction in metal thin films is caused by micro-plasticity arising from dislocation motions. Choi and Nix [5] summarized that internal friction is closely affected by amplitude, frequency, and temperature and film thickness. In these works, the thickness of the metal films ranged in micron and sub-micron scales. As a result, it leaves a gap in understanding the internal friction of ultra-thin Al films.

The goal of the present research is to investigate the dynamic properties of ultra-thin Al films (nanometer scale) at room temperature under high vacuum conditions. After resonant excitation, energy loss was estimated according to the decay in the oscillation amplitude of a vibrating structure. The effects of thickness and microstructure (grain size) were examined to determine the energy loss in thin metal films. While maintaining the same thickness, the grain size of the metal films was thoroughly controlled by annealing temperature and by altering the deposition process. The film thickness and grain size were observed because they exert the most significant influence on the dynamic properties of Al films used in MEMS and IC applications.

## 2. Methods and Experimental Design

### 2.1. Sample Design and Fabrication

The specimen included a cantilever beam made using silicon under uniform stress, carrying a comparatively large paddle plate (see Figure 1a). Sample dimensions were as follows: frame (20 mm × 20 mm), paddle plate (5 mm × 5 mm), and length of tapered beam (3 mm) varying in width (3.3 mm wide at the root narrowing to 1.5 mm where it connects to the paddle plate), as shown in Figure 1b. According to [7,12,16,17,18,19], standard etching techniques and Si IC prototyping were used to fabricate crystalline silicon with highly conductive properties. Figure 2 presents a cross-section of the chip with measurements of the constituent parts.

The process of fabrication is presented in Figure 3; a paddle-shaped cantilever beam with uniform stress distribution was used to support extremely thin Al films. The Si wafer was fabricated with a thickness of 250 µm, and the chip frame and paddle plate (forming in the middle of the chip frame) had the same thickness of 250 µm; the thickness of the tapered cantilever beam was 40 µm. Because of the stiffness between the frame, paddle plate, and tapered beam, all bending in the assembly occurred at the thin tapered cantilever beam. In the course of this research, the average- or low-conductivity wafers commonly used in the semiconductor industry were replaced with highly-conductive Si wafers. General Si chips are unable to obtain information directly if they do not have a conducting layer on the upper surface. The high conductivity of the Si chip enabled precise measurements without the need for additional components. The upper surface of the Si chip was covered by an Al film and the difference in the deformation of the paddle cantilever, without and with the Al film, provided the measurement information of the Al film. The thickness of the cantilever beam was much larger than that of thin Al films; consequently, due to beam deflection, the film experienced a uniform strain across its thickness which was seemingly equal to the strain on the top of the silicon surface.

Al films were deposited using a Pulse DC sputtering system at the Instrument Technology Research Center, Taiwan. A deposition pressure of 5 ± 10^−3^ torr and a deposition power of 200 W was maintained. This study obtained internal friction values within Al films of four different thicknesses. Al films of different thicknesses (as controlled by deposition time) were deposited on the top surface of the paddle samples. The thicknesses of the four Al films were 30, 68, 110, and 289 nm. Throughout the study, we carefully controlled the sputtering parameters and used details of the processing factors to control the grain size and film thickness. We could then confirm the film structure and grain size via SEM [20].

Previous research indicated that residual stress relaxation time in the evolution of ultra-thin films follows an exponential law on the order of tens of seconds [19]. Therefore, to avoid the effects of residual stress, all samples were held in the air, at room temperature, under constant humidity for at least three days following the deposition of Al to allow time for the residual stress resulting from the deposition process to relax to a steady-state. The film thicknesses of all Si chips were accurately measured by alpha-step.

### 2.2. System Design

The test system used in this work was a personal computer (PC), with NI LabVIEW software, a lock-in amplifier, charge sensitive preamplifier, power amplifier, BNC connecter, function generators, and data acquisition hardware. A schematic diagram of the experimental system is shown in Figure 4. A sine wave signal was sent by the PC to the BNC connecter. The signal was then processed and sent to the power amplifier by the BNC connecter during the experiment. The bending of the paddle occurred when the voltage at the deflection electrode was changed, which also caused a displacement current through the paddle capacitor and caused the total displacement current to change accordingly. The existence of two capacitors (reference capacitor and paddle capacitor) resulted in two displacement currents. The displacement current through the reference capacitor was maintained at a constant value throughout the experiment, while the other displacement current depended on the position of the paddle plate, which was determined by the capacitance of the paddle. The input of the lock-in amplifier with a mixed-signal (a 100 kHz sine wave modulated by the free vibration frequency) and a 100 kHz reference signal from a function generator (used to demodulate the mixed-signal) was used for the direct measurement of changes in the paddle capacitance. Using the rapid Fourier transform (FFT) method, the obtained results were converted from the time domain to the frequency domain.

### 2.3. Experimental Procedure

Underneath the paddle plate, a waveform was used to drive the electrode from where the excitation exerts the bending force. Excitation was applied using two different types of voltage: sweep frequency voltage and constant frequency voltage. The resonance of the paddle was measured using the sweep frequency excitation method. In contrast, constant frequency excitation was used to establish the amplitude of stability. After excitation was ceased, before measuring the free vibration of the paddle sample, sweep frequency as well as constant frequency excitation was used to measure the resonance of the paddles and establishing a staple amplitude, simultaneously. Free decay is commonly used to measure energy loss and internal friction. When the paddle specimen bent due to excitation, a corresponding change in the displacement current was registered in the coupling capacitor. The changes in the displacement current were immediately detected by the lock-in amplifier, which were recorded and stored using LabVIEW. The output signal from the lock-in amplifier was used to plot the output voltage against the experiment time to observe the response of the paddles. The frequency component of the response was analyzed by fast Fourier transform. The ambient pressure was an essential factor in the decay rate experiments; consequently, all capacitance measurements were performed under a high vacuum. In addition to measuring the decay rate of metal films alone, it was also necessary to measure the total decay rate reaction of the Si paddles with a thin metal film on the surface. Therefore, the decay rate of the Si paddles first needed be determined to distinguish between metal and Si segments. The results of free decay of the paddle samples with Al thin film (110 nm) on the surface are shown in Figure 5. Internal friction was calculated and determined based on the decreasing logarithm according to δ [21], which is dependent on the related data decay and frequency. The definition of logarithmic reduction is defined as follows:(1)$$\delta =\frac{\Delta W}{2W}$$
where Δ*W* is the energy dissipated in each cycle of oscillation and *W* is the vibration energy of the system. The experiment used a beam vibrating cantilever with two layers; the thin metal film was deposited on one face of the silicon beam. The internal friction in the metal film alone was derived from the total reaction as follows:(2)$$Q_{f}^{-1}=\frac{t_{Si}}{3t_{f}}\frac{E_{Si}}{E_{f}}\left(Q_{c}^{-1}-Q_{Si}^{-1}\right)$$
where $Q_{f}^{-1}$, $Q_{c}^{-1}$, $Q_{Si}^{-1}$ represents internal friction in the thin metal film, the internal friction in the Al/Si component, and the internal friction of Si alone, which is the weighted average of the internal friction associated with the thickness of the substrate ($Q_{Si}^{-1}$) and the film ($Q_{f}^{-1}$). The Young’s modulus of the Si substrate and Al film are presented by $E_{Si}$ (GPa) and $E_{f}$ (GPa), respectively. $t_{Si}$ (µm) is the thicknesses of the Si beam, while $t_{f}$ (nm) represents the thickness of the Al film. The pressure parameters in the measuring system were adjusted to the same values. This adjustment was made to cancel out the effects of gas damping in the following calculations.

## 3. Results and Discussion

### 3.1. Test Results of Al and Si Composite

Figure 6 illustrates the free decay results of Al film deposited on four Si paddles with different thickness. The Al film with a thickness of 289 nm coated on the paddles with pure Si was used to measure the free decay reaction. Comparing the decay rate of a pure Si paddle with that of the 289 nm Al thickness coated ones, a shift from 0.00561 to 0.0521 s^−1^ can be observed, leading to an increase in the logarithmic decrease from 2.2 × 10^−5^ to 2.03 × 10^−4^. The result infers that the energy stored inside a bare Si paddle decays ten times slower than that of a paddle deposited with Al with a thickness of 289 nm, caused by the internal energy loss in the Al layer. For an Al thickness of 110 nm, this decay rate shows an increase from 0.0049 s^−1^ to 0.0334 s^−1^, subsequently changing the logarithmic decrease from 1.93 × 10^−5^ to 1.31 × 10^−4^. Despite this fact that the resonance frequency film of thicknesses less than 100 nm cannot change to a demonstrable degree, both the thicknesses of 70 and 30 nm Al coated film showed an increase in the decay rates at which the 70 nm Al film decay rate increases from 0.00509 to 0.0224 s^−1^, and 0.00441 to 0.031 s^−1^ for the 30 nm Al film. These decay rate increases also lead to a shift in logarithmic decrease from 2.18 × 10^−5^ to 9.58 × 10^−5^, and from 1.8 × 10^−5^ to 5.26 × 10^−5^ for 70 and 30 nm, respectively. In the above results, the decay rate was greatly influenced by the film thickness; when the thickness of the thin film decreased, the decay rate also decreased, and vice versa. Nonetheless, it can be observed that the decay rate varied significantly (about 3 times) even in the sample with an Al film of only 30 nm in thickness. Furthermore, the decrease in the film/substrate composite relative to the paddle (bare Si) was greater than the logarithmic decrease. This suggests that even if the film thickness is only a few percent (0.08 to 0.8%) of the total thickness, the damping of the composite is essential. The energy loss mechanism is detailed in the following section.

### 3.2. Energy Loss Analysis of Pure Al Films

The aim of this study was to determine how the thickness of pure metal films influence internal friction. Internal friction in pure Al films was derived from the total response using Equation (2), with Young’s moduli of Si and Al as 127 and 76 GPa, respectively (using microtensile testing) [22,23,24]. A total of ten specimens were tested for each film thickness for the internal friction measurement. The average internal friction in the Al films was calculated as follows: 4.68 × 10^−3^ (289 nm), 7.55 × 10^−3^ (110 nm), 7.72 × 10^−3^ (68 nm), and 8.39 × 10^−3^ (30 nm). Figure 7 plots the internal friction the pure Al film against the deposited thickness, in which the internal friction decreased with an increase in film thickness. As shown in the figure, internal friction decreased almost linearly with an increase in film thickness. Internal friction decreased from 8.39 × 10^−3^ to 4.68 × 10^−3^ as the thickness of the film was increased from 30 to 289 nm. These results indicated that the internal friction in ultra-thin Al films of less than 0.3 μm decreased with an increase in film thickness.

Figure 8 shows the SEM plan-view images of each sample, the Al films on the sample surface thickness vary from 30 to 289 nm. The micrographic structure in all four thickness films is difficult to vary and no significant differences were observed in the grain sizes (~30 to 50 nm) between these four samples.

### 3.3. Energy Loss Analysis of Pure Al Films with Annealing

Al films in four thicknesses less than 300 nm were annealed for 400 min (not including the temperature ramp-down time) in an atmospheric furnace (REX-P200, RKC, Tokyo, Japan) with the annealing temperature set at 280 °C (furnace temperature was increased or decreased 5 °C per min). To avoid Al film surface oxidation, 200 sccm of forming gas (N_2_/H_2_ = 9) was injected into the furnace tube. A comparison was then conducted with the results from the as-deposited films.

The decay rate of the annealed paddles differed slightly from that of paddles that did not undergo annealing. The internal friction in the annealed Al films was calculated as 8.43 × 10^−3^ (30 nm), 9.94 × 10^−3^ (68 nm), and 1.09 × 10^−2^ (110 nm). Figure 9 presents the relationship between film thickness and the internal friction in the annealed Al films. Annealed Al films presented a greater internal friction than non-annealed Al films when film thickness was increased.

These results indicate that the internal friction in the annealed Al films was higher than that in the as-deposited films with thicknesses less than 110 nm. However, in the 289 nm Al film, the internal friction decreased from 4.89 × 10^−3^ (as-deposited) to 4.03 × 10^−3^ (annealed), which is lower than that of the as-deposited film.

## 4. Discussion

The results contained in this paper summarize the measurements of the effects of frequency and thickness on internal friction in ultra-thin nano-scale Al films. The results shown direction and immediate information for the design of high frequency MEMS/NEMS RF resonators. Accordingly, annealing Al thin layer coatings has proven to be an effective process for reducing dissipation in layered resonators [7,25]. Therefore, the process versus structure for energy dissipation relationships can be guided for the design of a variety of resonant device applications for MEMS.

Considering the magnitude and source of energy loss in nanocrystalline Al films from 30 to 289 nm, the internal friction ranged from 4.68 × 10^−3^ to 8.93 × 10^−3^ at room temperature for the films measured in this study, comparing these values with previous measurements is not feasible because of the many differences in grain size, film thickness, frequency, processing methods, annealing conditions, temperature and measurement methods. Therefore, an in-depth discussing as follows.

Energy loss or internal friction can be attributed to several sources: (a) thermoelastic damping; (b) sliding at the cap/metal/Si interface; (c) surface defects and (d) crystallographic defects within the film. A previous study [24] found the contributions of Al thin film to thermoelastic damping can be negligible. The (b) and (c) sources can also be ruled out based on our study. If interfacial sliding or surface defect energy dissipation were the major sources, then damping in the Al/Si bilayers must be independent of Al film thickness [11]. However, our measurements consistently showed that damping in Al/Si bilayers increased in proportion to the thickness of the Al film for the full range of frequencies and film thicknesses. A similar behavior was found in another study using a silicon microcantilever platform for a wider range of thicknesses but over a smaller frequency spectrum [7].

Another reason interfacial sliding can be ruled out is because it can always maintain excellent adhesion between Al/Si interfaces from the standard microfabrication processes for the tested samples. Moreover, the geometric design of an ultrathin Al film on a thick substrate suggests that the shear stress at the Al/Si interface can be negligible.

In regard to sliding at the cap/metal, forming gas was used to avoid surface oxidization of the Al films on the annealing paddles within an air environment during this study. The forming gas eliminated surface oxidation; however, it may have led to the growth of a thin nitride layer on the Al film surface. Energy dispersive spectrometry (EDS) was used to analyze the surface of Al thin films without and with annealing. Figure 10a,b presents the EDS results in the presence of elements from the Al thin films, without and with annealing, respectively. As shown in Figure 10a, the Al film without annealing presented peaks indicating the existence of only Si and Al with a trace of elemental O. The Si and O can be attributed to the substrate and the native oxide layer. The spectrum did not reveal any significant differences in the samples annealed at 280 °C for 400 min, compared with the non-annealed films. Nonetheless, the weight percentage of O presented a slight increase after annealing. The EDS results indicate that the surface of the Al films did not produce a layer of aluminum nitride after annealing; instead, a slight but negligible increase was observed in the thickness of the aluminum oxide layer.

By taking above mentioned factors into account, and including a process of elimination, it leads into the conclusion that the energy loss and internal friction of ultrathin Al film is dominated by the motion of crystallographic defects within the Al film.

There are several types of crystallographic defects that exist within Al thin films. The results shown in Figure 7 imply that there is a link between grain boundaries and energy dissipation. As shown in Figure 7, the internal friction increased with a decrease in the thickness of Al films below 200 nm at room temperature. Previously, Berry suggested that the internal friction in thin Al films at room temperature (300 K) could be attributed to short-range relaxation associated with grain boundary sliding [26]. Nix et al. presented dynamic measurement results related to damping in thin metal films [27]. They measured the internal friction in Al films at three different frequencies over a range of temperatures to calculate the activation energy. Their results suggested that the damping mechanism was grain boundary sliding controlled by grain boundary diffusion.

Furthermore, the results shown in Figure 9 suggest that with an Al film thickness over 200 nm, microstructural changes caused by annealing can increase the median grain size and broaden the grain size distribution. These will reduce the total grain boundary area and result in a drastic reduction in internal friction, as shown in the film thickness of 289 nm. This link has been noted in previous measurements of internal friction as a function of temperature in Al films [25,26,27,28,29].

In contrast, grain boundaries at a distance from the interface underwent regular sliding when the structure was vibrated. Decreasing the percentage of constricted grain boundaries promoted a thickness-dependent increase in the energy loss of Al films. These results are consistent with others [7,29,30] indicating that the grain sizes and grain boundary density strongly affect the internal friction and energy loss of the films.

Kê [31,32] studied the internal friction in polycrystalline and single crystal aluminum. Their results indicated that the internal friction in annealed polycrystalline aluminum is due to sliding along the grain boundaries. It was proposed that if the energy stored in the bare Si paddle samples could not be dissipated through grain boundary sliding, the resulting internal friction would be less than that of the paddle samples with Al film.

With the Al film thickness less than 100 nm, in Figure 9, the internal friction in the annealed Al films presented a greater internal friction than non-annealed Al films when film the thickness was increased. Previously, it was found that with the Al film thicknesses of 100 nm, there was little grain growth at relatively low annealing temperatures (100–350 °C) [7]. Moreover, no significant difference was observed in the microstructure between annealed and non-annealed Al thin film samples [33]. In addition, this indicated that these films contained a broad distribution of grain sizes and few grains grew. This could explain why there is no reduction in internal friction for annealed Al films with thicknesses of 38 nm to 110 nm. The increasing energy dissipation for these annealed Al films could be the results of strain in Al thin film changes due to annealing. Past studies [34] indicated that annealed Al film will change the strain. In addition, Nishino et al. claimed that the reduction in the relaxation peak following a decrease in film thickness could be attributed to the constraining effects of a rigid substrate [12]. Further study of residual strain measurements will be performed using FIB-DIC and the results will be presented in a future paper.

## 5. Conclusions

This study proposed a novel approach to the measurement of energy loss and internal friction in Al thin films. Under high vacuum conditions, we employed a test specimen with uniform stress distribution to correlate capacitance measurements. The experiments produced usable data for thin Al films less than 100 nm in thickness. Film thickness demonstrated a solid relationship with internal friction and energy loss in metal films. For Al films deposited and measured at room temperature, the thickness of the films is inversely proportional to the internal friction. We also examined the internal friction of annealed Al films in terms of microstructure. There is no doubt that the microstructural features of the films were the principal factor contributing to internal friction. In fact, our results all prove that grain boundaries are the leading cause of internal friction, as a result of grain boundary sliding.

## Figures and Tables

**Figure 1 materials-14-03401-f001:**
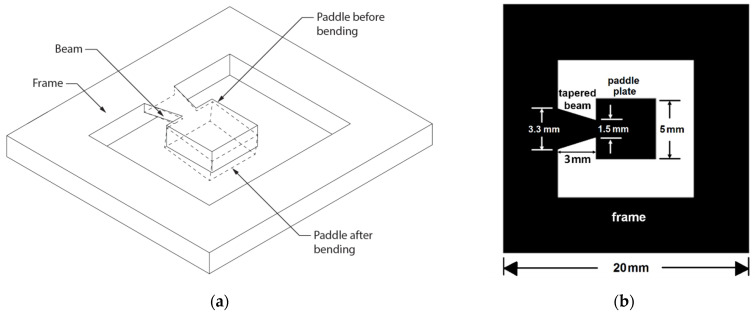
(**a**) A schematic of the mechanism. (**b**) Configuration of a rectangular paddle micro-cantilever.

**Figure 2 materials-14-03401-f002:**
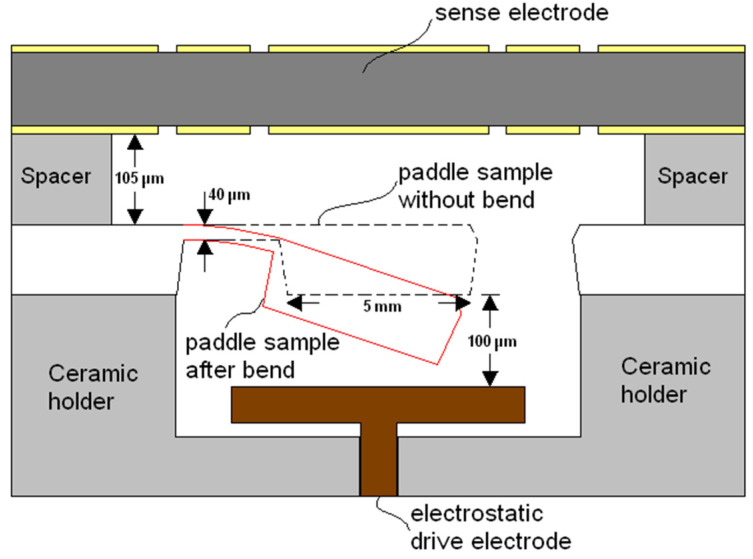
Cross-section of the chip with measurements of the constituent parts.

**Figure 3 materials-14-03401-f003:**
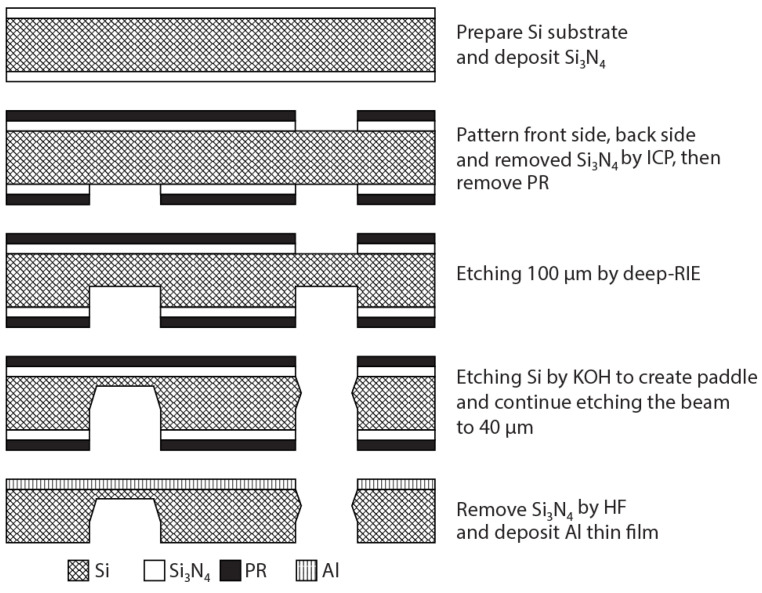
The fabrication process of the paddle micro-cantilever.

**Figure 4 materials-14-03401-f004:**
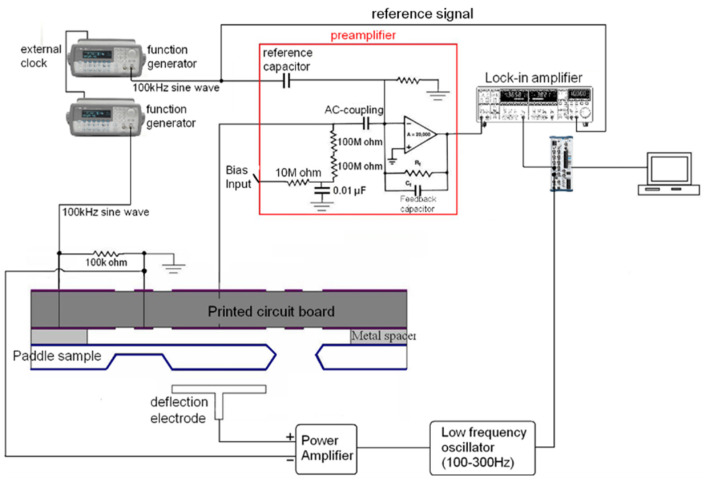
System and principle of operation for measuring energy loss.

**Figure 5 materials-14-03401-f005:**
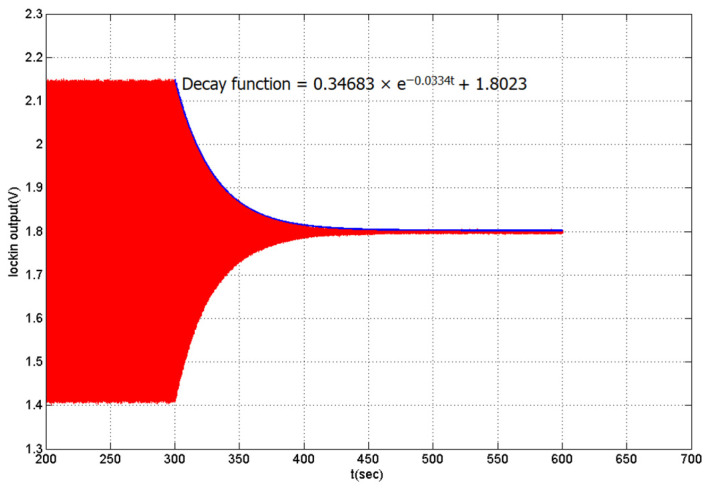
Free damping response of the 110 nm Al film on surface paddle samples.

**Figure 6 materials-14-03401-f006:**
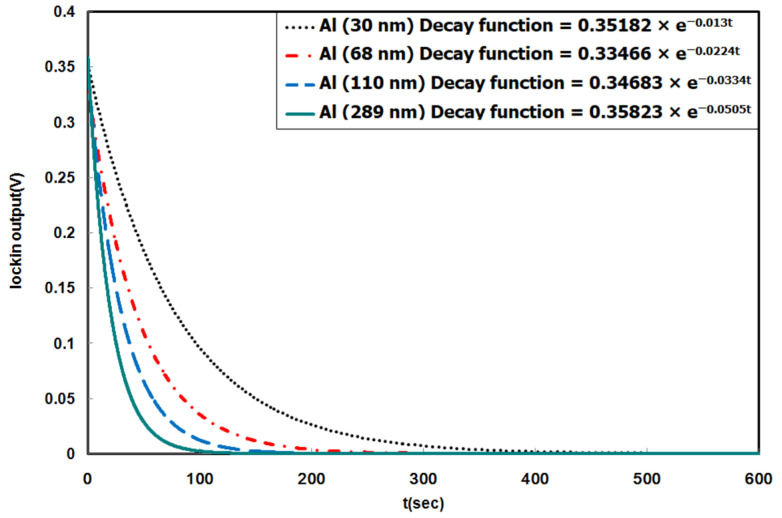
Comparison of the free decay with different thicknesses of Al films 30, 68, 110, and 289 nm.

**Figure 7 materials-14-03401-f007:**
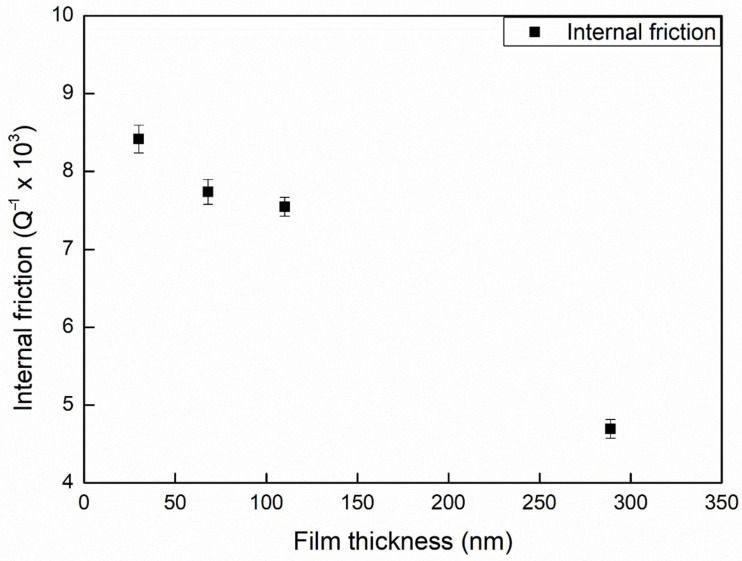
The results of the internal frictional comparison while the thickness of the Al film was increased.

**Figure 8 materials-14-03401-f008:**
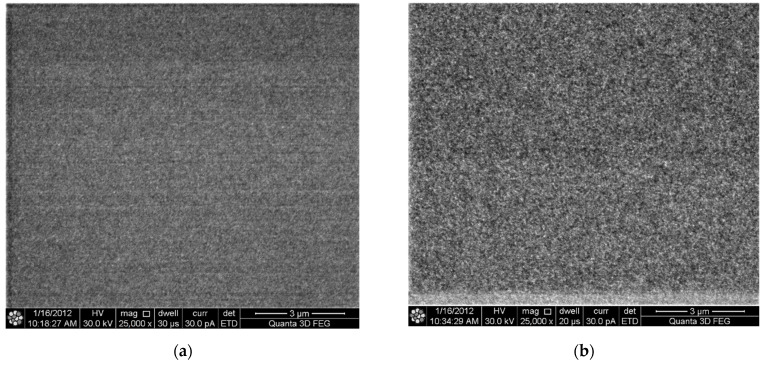
SEM image for Al film surface (**a**) 30 nm; (**b**) 68 nm; (**c**) 110 nm; (**d**) 289 nm.

**Figure 9 materials-14-03401-f009:**
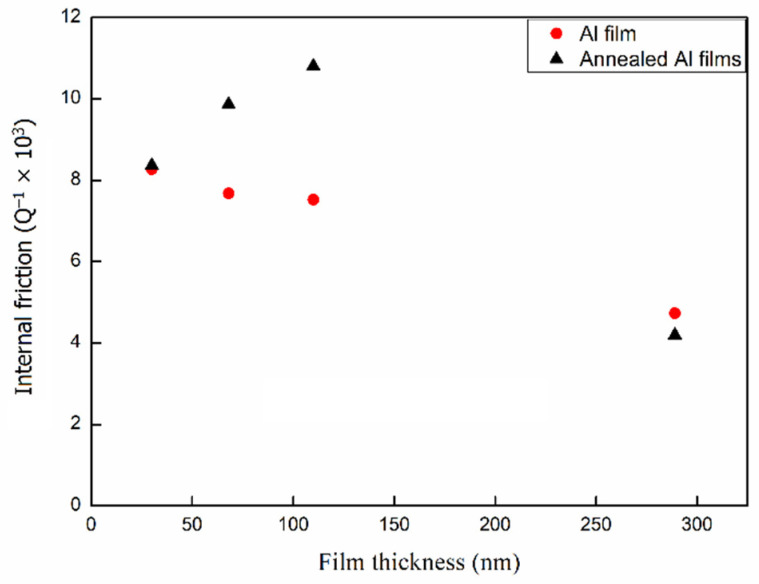
Results comparing internal friction between Al annealed and unannealed membranes.

**Figure 10 materials-14-03401-f010:**
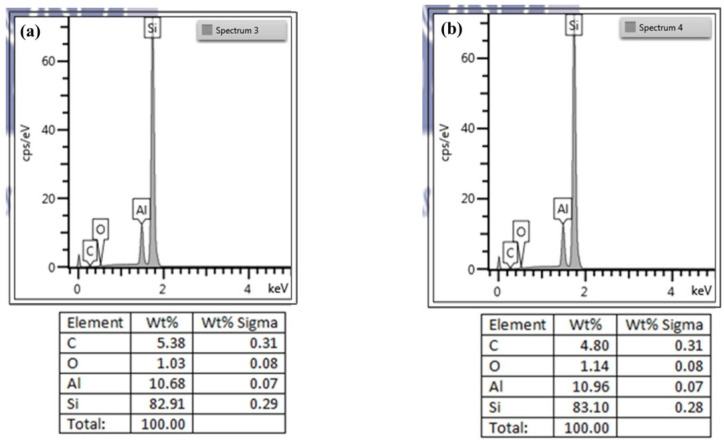
The result used EDS to analyze the surface of Al thin films. (**a**) Unannealed Al thin films. (**b**) Annealed Al thin films.

## Data Availability

Data is contained within the article.

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
