# Peer review of "Size Effects in Internal Friction of Nanocrystalline Aluminum Films"

_materials, 2021, doi:10.3390/ma14123401_

Round 1

Reviewer 1 Report

The subject of this work is interesting, however it can not be accepted in its present form due to the following major corrections required:

  1. The manuscript is more technical than scientific. The whole paper should be rewritten carefully. Objectives and vision are not clear, the link between other published works and this work is missed, the starting point (ignition point) shoul be more clear in tis work. I mean from where you start and why
  2. discussions (interpretations and explanations) should be more indeep. Speculative discussions is not helpful. The authours should be care of this imporatnt section. Indeep scientific texts should be added ionto this section with more references. 

Author Response

Response to Reviewer 1 Comments

The manuscript is more technical than scientific. The whole paper should be rewritten carefully. Objectives and vision are not clear, the link between other published works and this work is missed, the starting point (ignition point) should be more clear in this work. I mean from where you start and why

Response: Thank you very much for your review together with very important suggestions. We had carefully taken your suggestions and updated in this submission.

Introduction:

MEMS devices are electrically controlled, but they are susceptible to mechanical failure and damping due to internal friction which is created by the impact between thin metal films on dynamic loads. In a micrometer scale or nanometer scale, measuring the internal friction of a thin film could help determine the relaxation processes involved in boundary diffusion and sliding [5-7]. Prieler et al. [8] and Illés et al. [9] performed a systematic investigation of the relaxation of thin Al films on a Si substrate, in which the grain size was completely controlled and varied solely with film thickness [8-9]. There results show that the grain boundary sliding and diffusion are related to the internal friction. In a different study, the internal friction between the free-standing membranes and the films attached to the substrate was compared [10-11]. The authors concluded that because the substrate restricts grain boundary sliding near the interface, the internal friction in free-standing films is greater than that in the film attached to a substrate.

In recent study, Fujiwara et al. [12] found internal friction in thin films was due to grain boundary anelastic processes. Nishino [13–15] illustrated that the dependence of amplitude for the internal friction in metal thin films is caused by micro-plasticity arising from dislocation motion. Choi and Nix [5] summarized that internal friction is closely affected by the amplitude, frequency, and temperature and film thickness. In these work, the thickness of the metal films were ranged at the micron and sub-micron scales. As a result, it leaves a gap for understanding the internal friction of ultra-thin Al films.

The goal of the present research is to investigate the dynamic properties of ultra-thin Al films (nanometer scale) at room temperature under high vacuum conditions. After resonant excitation, energy loss was estimated according to the decay in the oscillation amplitude of a vibrating structure. The effect of thickness and microstructure (grain size) were examined to determine the energy loss in thin metal films. While maintaining the same thickness, the grain size of the metal films was thoroughly controlled by annealing temperature and by altering the deposition process. The film thickness and grain size were observed because they exert the most significant influence on the dynamic properties of the Al films used in MEMS and IC applications.

Discussions (interpretations and explanations) should be more in deep. Speculative discussions are not helpful. The authors should be careful of this important section. Deep scientific texts should be added to this section with more references.

Response: We had add an addition of discussions and updated in this submission.

  1. Discussion

The results contained in this paper summarize the measurements of the effects of fre-quency and thickness on internal friction in ultra-thin nano-scale Al films. It has the direct and immediate information for the design of high frequency MEMS/NEMS RF resonators. Accordingly, annealing Al thin layer coatings has proven to be an effective process for reducing dissipation in layered resonators [16, 26]. Therefore, the process versus structure for energy dissipation relationships can be guided for the design of a variety of resonant devices application for MEMS.

Now consider the magnitude and source of energy loss in nanocrystalline Al films from 30 nm to 289 nm. Internal friction ranges from 4.68 × 10−3 to 8.93 × 10−3 at room temperature for the films measured in this study. Comparing these values with previous measurement is not feasible because of many differences in grain size, film thickness, fre-quency, processing methods, annealing conditions, temperature and measurement methods. Therefore, in depth discussion is as following.

Energy loss or internal friction can be attributed to several sources: (a) thermoelastic damping; (b) sliding at the cap/metal/Si interface; (c) surface defects and (d) crystallo-graphic defects within the film. Previous study [26] found the contributions of Al thin film to thermoelastic damping can be negligible. The (b) and (c) sources can also be ruled out based on our study. If interfacial sliding or surface defect energy dissipation were the major sources, then the damping in Al/Si bilayers must be independent of Al film thick-ness [11]. However, our measurements consistently show that the damping in Al/Si bi-layers increases in proportion to the thickness of the Al film for the full range of frequency and film thickness. Similar behaviour was found in another study using the silicon microcantilever platform for a wider range of thickness over a smaller frequency spectrum [16].

Another reason interfacial sliding can be ruled out because it can always maintain excellent adhesion between Al/Si interfaces from the well taken standard microfabrication processes for the tested samples. Moreover, the geometrical design of an ultrathin Al film on a thick substrate suggests that the shear stress at the Al/Si interface can be negligible.

In regarding sliding at the cap/metal, forming gas was used to avoid the surface oxidization of Al films on the annealing paddles within an air environment during this study. The forming gas eliminated surface oxidation; however, it may have led to the growth of a thin nitride layer on the Al film surface. Energy dispersive spectrometry (EDS) was used to analyse the surface of the Al thin films without and with annealing. Figures 10(a) and 10(b) present EDS results in the presence of elements from Al thin films without and with annealing, respectively. As shown in Fig. 10(a), the Al film without annealing presented peaks indicating the existence of only Si and Al with a trace of elemental O. The Si and O can be attributed to the substrate and native oxide layer. The spectrum did not reveal any significant differences in samples annealed at 280oC for 400 minutes, compared with the non-annealed films. Nonetheless, the weight percentage of O presented a slight increase after annealing. EDS results indicate that the surface of Al films did not produce a layer of aluminium nitride after annealing; instead, a slight increase, but negligible, was observed in the thickness of the aluminium oxide layer.

By taking above mention factors into account, and includes a process of elimination, it leads into the conclusion that the energy loss and internal friction of ultrathin Al film is dominated by the motion of crystallographic defects within the Al film.

There are several types of crystallographic defects exist within Al thin films. The result shown in figure 7 implies that there is a link between grain boundaries and energy dis-sipation. As shown in figure 7, internal friction increased with a decrease in the thickness of Al films below 200 nm at room temperature. According to the previous studies on the microstructure of ultra-thin Al film [16, 21] grain size increased with an increase in film thickness. These larger grains reduced the number of grain boundaries, thereby limiting the source of grain boundary sliding and diffusion. Previously, Berry suggested that internal friction in thin Al films at room temperature (300K) could be attributed to short-range relaxation associated with grain boundary sliding [27]. Nix et al. presented dynamic measurement results related to damping in thin metal films [28]. They measured the internal friction in Al films at three different frequencies over a range of temperatures to calculate the activation energy. Their results suggested that the damping mechanism was grain boundary sliding controlled by grain boundary diffusion.

Furthermore, the results shown in figure 9 suggest that with the Al film thickness over 200 nm, the microstructural changes caused by annealing can increase the median grain size and broaden of the grain size distribution. These will reduce the total grain boundary area and result in a drastic reduction in internal friction as shown in the film thickness of 289 nm. This link has been noted in previous measurements of internal friction as a function of temperature in Al films [26–30].

In contrast, grain boundaries at a distance from the interfaces underwent regular sliding when the structure was vibrated. Decreasing the percentage of constricted grain boundaries promoted a thickness-dependent increase in the energy loss of Al films. These results show consistent with others [7, 30, 31] indicating that the grain sizes and grain boundary density strongly affect the internal friction and the energy loss of the films.

Kê [11] studied the internal friction in polycrystalline and single crystal aluminium. Their results indicated that the internal friction in annealed polycrystalline aluminium is due to sliding along the grain boundaries. It is proposed that if the energy stored in the bare Si paddle samples could not be dissipated through grain boundary sliding, the resulting internal friction would be less than that of the paddle samples with Al film.  

With the Al film thickness less than 100 nm in figure 9, the internal friction in the an-nealed Al films presented a bit greater internal friction than non-annealed Al films when film thickness was increased. Previously, it was found that with the Al film thicknesses of 100 nm, there was little grain growth at relatively low annealing temperatures (100–350◦C) [16]. Moreover, no significant difference was observed in the microstructure between annealed and non-annealed Al thin film samples [32]. In addition, they indicated these films contained a broad distribution of grain sizes and a few grains grew. This could explain why there is no reduction in internal friction for annealed Al films with thickness of 38 nm to 110 nm. The increasing energy dissipation for these annealed Al films could be the results of strain in Al thin film changes due to annealing. Past studies [33] indicated that the annealed Al film will change the strain. In addition, Nishino et al. claimed that the reduction in relaxation peak following a decrease in film thickness could be attributed to the constraining effects of the rigid substrate [24]. Further study on the residual strain measurement will perform using FIB-DIC and the results will present in the future paper.

Reviewer 2 Report

Authors have reported the effects of the characteristics of MEMS materials i.e. Al thin film. They have investigated the dynamic properties of ultra-thin Al film (nanometer scale) deposited under high vacuum. It is claimed that the internal friction of ultra-thin and thin Al films is more dependent on the grain boundaries than the film thickness.

It is recommended to consider after addressing the following comments.

  1. Include error bar in Fig. 7.
  2. Why there are more white granular features in Fig. 8C?
  3. In Fig. 9, is the fourth point for annealed film out of range?
  4. It has mentioned that “These results show consistant ( note I get confused with this word) with others [7, 25, 26] indicating that the grain sizes and grain boundary density strongly affect the internal friction and the energy loss of the films.” If so, do you have an idea of how to minimize it?

Author Response

Response to Reviewer 2 Comments

Authors have reported the effects of the characteristics of MEMS materials i.e. Al thin film. They have investigated the dynamic properties of ultra-thin Al film (nanometer scale) deposited under high vacuum. It is claimed that the internal friction of ultra-thin and thin Al films is more dependent on the grain boundaries than the film thickness. It is recommended to consider after addressing the following comments.

 Response: Thank you very much for your review together with very positive suggestions. We had carefully taken your suggestions and updated in this submission. We hope it is satisfied.

Point 1: Include error bar in Fig. 7.

Response 1: We have revised the Figure7 to add the error bar. That shall provide the clear information in the figures. In the text, we also revised “Total of ten specimens was tested for each film thicknesses for the internal friction measurement. The average internal friction in the Al films was calculated as follows:” We hope it is satisfied.

Point 2: Why there are more white granular features in Fig. 8C?

Response 2: We had checked again and believed the more white granular features in this figure was due to higher contrast setting of the image and this should not be the issue in comparison with others.

Point 3: In Fig. 9, is the fourth point for annealed film out of range?

Response 3: We have corrected in the revision of manuscript.

Point 4: It has mentioned that “These results show consistent (note I get confused with this word) with others [7, 25, 26] indicating that the grain sizes and grain boundary density strongly affect the internal friction and the energy loss of the films.” If so, do you have an idea of how to minimize it?

Response 4: The change of grain boundaries in the microstructure will affect the internal energy consumption of the film. Comparing the results before and after annealing, the grains are smaller and more numerous before annealing, and the number of grain boundaries are larger as indicated results from others [7, 25, 26]. If the film thickness is thicker than 200 nm, the grain size grows much larger after annealing, and the area of grain boundary is reduced due to the larger grain size, and the relative internal energy consumption will be reduced since the area of grain boundary is reduced. To minimize it, either deposition process or the annealing procedure can be used to control it. We had updated in the revised manuscript.

Reviewer 3 Report

The manuscript is devoted to study the internal friction of very thin Al film that is in particularly important for MEMS devices. The experimental design for study mechanical properties and obtained results for internal friction are very interesting. However, in discussing the results, the authors rely on the results of structural analysis that do not exist in manuscript  although they are in Abstract and in Introduction. 

There are technical weaknesses in presentation ( the unclear sentences and some  graphical presentations should be improved).

18 “Grain size and film thickness size were strictly controlled considering the quasi-static properties of the films.”

How?

 19 “This study found that the internal friction of ultra-thin and thin Al films was more dependent on the grain boundaries than the film thickness.”

This result is not present in the manuscript.

“The effect of thickness and microstructure (grain size) were examined to determine the energy loss in thin metal films. While maintaining the same thickness, the grain size of the metal films was thoroughly controlled by annealing temperature and by altering the deposition process. The film thickness and grain size were measured because they exert the most significant influence on  the dynamic properties of the films used in MEMS and IC applications.”

I do not see how the film structure was determined and what are the results.  In particularly what are the grain sizes?

200  “According to observations of microstructure by ionic beam scanning, grain size increased with an increase in film thickness.”  Measured in this work or in some other?

Fig 8

It is not much seen from presented micrographs. Maybe larger magnification could help?

213 “The grain structure in all 4 Al films is difficult to vary, however, the grain size in all cases is smaller than for the Cu films”

What authors wanted to say?

Fig 9

Discussion of results in Fig 9 is not convincing. Why is dependence of internal friction on thickness different for as deposited and annealed films?

  (228 “However, no significant difference was observed in the microstructure between annealed and non-annealed samples”)

Maybe strain in Al thin film changes due to annealing?

244 “Additionally, this study used electron spectroscopy for chemical analysis (ESCA) to provide depth profiles  of the Al thin films.”

It would be interesting to see ESCA results-

2467 “We determined that the grain boundaries in the Al films produced sliding near the Al2O3/Al and Al/Si interfaces.”

How?

250 “Decreasing the percentage of constricted grain boundaries promoted a thickness-dependent increase in the energy loss of Al films. These results show consistant with others [7, 25, 26] indicating that the grain sizes and grain boundary density  strongly affect the internal friction and the energy loss of the films.”

This part of discussion is not clear.

Fig 10 should be redesigned. Maybe, the energy range up to 4 keV would be enough; the number on the axes should be the same size as on the other graphs and the size of letters in accompanied table should be the same size as in text

What is presented on micrographs in Fig 10? In my opinion, they could be omitted.

Author Response

Response to Reviewer 3 Comments

reviewer 3: The manuscript is devoted to study the internal friction of very thin Al film that is in particularly important for MEMS devices. The experimental design for study mechanical properties and obtained results for internal friction are very interesting. However, in discussing the results, the authors rely on the results of structural analysis that do not exist in manuscript although they are in Abstract and in Introduction.

Response: Thank you very much for your review together with very important suggestions. We had carefully taken your suggestions and updated in this submission.

There are technical weaknesses in presentation (the unclear sentences and some graphical presentations should be improved).

Response: Thank you very much for your review together with very important suggestions. We had carefully taken your suggestions and updated in this submission.

Point 1: Page 1, line 18 “Grain size and film thickness size were strictly controlled considering the quasi-static properties of the films.”

How?

Response: In standard microfabrication processes, it can now maintain excellent deposition parameters for deposition of the tested samples. In this study, we carefully control sputtering parameters and using details processing factors to control the grain size and film thickness.

Point 2: Page 1, line 19 “This study found that the internal friction of ultra-thin and thin Al films was more dependent on the grain boundaries than the film thickness.”

This result is not present in the manuscript.

Response: We have corrected in the revision of manuscript.

Point 3: Page 2, line 58 “The effect of thickness and microstructure (grain size) were examined to determine the energy loss in thin metal films. While maintaining the same thickness, the grain size of the metal films was thoroughly controlled by annealing temperature and by altering the deposition process. The film thickness and grain size were measured because they exert the most significant influence on the dynamic properties of the films used in MEMS and IC applications.”

I do not see how the film structure was determined and what are the results.  In particularly what are the grain sizes?

Response: In standard microfabrication processes, it can now maintain excellent deposition parameters for deposition of the tested samples. In this study, we carefully control sputtering parameters and using details processing factors to control the grain size and film thickness. It then can confirm the film structure and grain size through SEM [23].

Point4: Page 9, line 200 “According to observations of microstructure by ionic beam scanning, grain size increased with an increase in film thickness.” Measured in this work or in some other?

Response: In this study, we carefully control sputtering parameters and using details processing factors to control the grain size and film thickness. It then can confirm the film structure and grain size through SEM [23].We had updated in this submission.

Fig 8. It is not much seen from presented micrographs. Maybe larger magnification could help?

Response: We had add an addition of discussions and updated in this submission.

Point 5: Page 9, line 213 “The grain structure in all 4 Al films is difficult to vary, however, the grain size in all cases is smaller than for the Cu films”

What authors wanted to say?

Response: We corrected the sentence and had add an addition of discussions and updated in this submission.

Point 6: Fig 9. Discussion of results in Fig 9 is not convincing. Why is dependence of internal friction on thickness different for as deposited and annealed films?

Response: We had add an addition of discussions and updated in this submission.

Point 7: Page 10, line 228 “However, no significant difference was observed in the microstructure between annealed and non-annealed samples”

Maybe strain in Al thin film changes due to annealing?

Response: You raised a very good point and this leads us to explain the behavior. The increasing energy dissipation for these annealed Al films could be the result of strain in Al thin film changes due to annealing. Past studies [33 ] indicated that the annealed Al film will change the strain. Further study on residuals strain measurement will perform using FIB-DIC and the results will present in the future paper.

  1. Doerner, M. F.; Brennan, S. "Strain distribution in thin aluminum films using x‐ray depth profiling." J. of applied physics 1988, 63.1, 126-131.

Point 8: Page 11, line 244 “Additionally, this study used electron spectroscopy for chemical analysis (ESCA) to provide depth profiles of the Al thin films.”

It would be interesting to see ESCA results-

Response: We did Energy dispersive spectrometry (EDS)  and it was used to analyze the surface of the Al thin films without and with annealing but not complete spectroscopy for chemical analysis (ESCA). In order to avoid confusion, we removed these sentences.  

Point 9: Page 11, line 246-7 “We determined that the grain boundaries in the Al films produced sliding near the Al2O3/Al and Al/Si interfaces.”

How?

Response: The reason interfacial sliding can be ruled out because it can always maintain excellent adhesion between Al/Si interfaces from the well taken standard microfabrication processes for the tested samples. Moreover, the geometrical design of an ultrathin Al film on a thick substrate suggests that the shear stress at the Al/Si interface can be negligible. We have revised it for this revision.

  Point 10: Page 11, line 250 “Decreasing the percentage of constricted grain boundaries promoted a thickness-dependent increase in the energy loss of Al films. These results show consistent with others [7, 25, 26] indicating that the grain sizes and grain boundary density strongly affect the internal friction and the energy loss of the films.”

This part of discussion is not clear.

Response 4: The change of grain boundaries in the microstructure will affect the internal energy consumption of the film. Comparing the results before and after annealing, the grains are smaller and more numerous before annealing, and the number of grain boundaries are larger as indicated results from others [7, 25, 26]. If the film thickness is thicker than 200 nm, the grain size grows much larger after annealing, and the area of grain boundary is reduced due to the larger grain size, and the relative internal energy consumption will be reduced since the area of grain boundary is reduced. To minimize it, either deposition process or the annealing procedure can be used to control it. We had updated in the revised manuscript.

Point 11:  Page 12, Fig 10 should be redesigned. Maybe, the energy range up to 4 keV would be enough; the number on the axes should be the same size as on the other graphs and the size of letters in accompanied table should be the same size as in text.

What is presented on micrographs in Fig 10? In my opinion, they could be omitted.

 Response: We modify the energy range to 4 keV and keep the table. We have revised Fig. 10 for this revision.

Round 2

Reviewer 1 Report

The manuscript can be accepted in its present form. The authours did most of required corrections.

Author Response

The manuscript can be accepted in its present form. The authours did most of required corrections.

Response: Thank you very much.

Reviewer 3 Report

The revised manuscript is considerably better than the original. But I still have two remarks

  • There is still no data on the structure of the deposited films.  What are the (expected if not measured) grain sizes?
  • 296 “Figures 8 shows the SEM plan-view images of each samples, the Al films on the sample surface thickness from 30 nm to 289 nm. The micrographic structure in all four thickness films is difficult to vary and no significant difference was observed in the grain sizes between these samples.”

In discussion, opposite to above finding authors claim that with increasing thickness, the grains are larger.

432 “According to the previous studies on the microstructure of ultra-thin Al film [16, 21] grain size increased with an increase in film thickness.”

Both statements cannot be in the same manuscript.

Author Response

The revised manuscript is considerably better than the original. But I still have two remarks

Response: Thank you very much for your review together with very important suggestions. We had carefully taken your suggestions and updated in this submission.

There is still no data on the structure of the deposited films.  What are the (expected if not measured) grain sizes?

Response: Thank you very much for your suggestions. Currently, we don’t have handful data on the structure of the deposited films. However, based on the information we had it is expected the grain sizes of all tested samples were from 30 nm to 50 nm. We had updated in this submission.

296 “Figures 8 shows the SEM plan-view images of each samples, the Al films on the sample surface thickness from 30 nm to 289 nm. The micrographic structure in all four thickness films is difficult to vary and no significant difference was observed in the grain sizes between these samples.”

In discussion, opposite to above finding authors claim that with increasing thickness, the grains are larger.

432 “According to the previous studies on the microstructure of ultra-thin Al film [16, 21] grain size increased with an increase in film thickness.”

Both statements cannot be in the same manuscript.

Response: Thank you very much for your very important suggestions. It is true that both statements cannot be together in this manuscript. We had carefully update it by removing 432 and revise the writing based on what we had studied in this work.